# MIRRORS ICG: Perfusion Assessment Using Indocyanine Green (ICG) Peritoneal Angiography during Robotic Interval Cytoreductive Surgery for Advanced Ovarian Cancer

**DOI:** 10.3390/cancers16152689

**Published:** 2024-07-29

**Authors:** Christina Uwins, Agnieszka Michael, Simon S. Skene, Hersha Patel, Patricia Ellis, Jayanta Chatterjee, Anil Tailor, Simon Butler-Manuel

**Affiliations:** 1School of Biosciences and Medicine, University of Surrey, Leggett Building, Daphne Jackson Road, Guildford GU2 7WG, UK; a.michael@surrey.ac.uk; 2Academic Department of Gynaecological Oncology, Royal Surrey NHS Foundation Trust, Egerton Road, Guildford GU2 7XX, UK; 3St Luke’s Cancer Centre, Royal Surrey NHS Foundation Trust, Guildford GU2 7XX, UK; 4Surrey Clinical Trials Unit, University of Surrey, Clinical Research Building, Egerton Road, Guildford GU2 7XP, UK; s.skene@surrey.ac.uk

**Keywords:** robotic surgery, ovarian cancer, interval, cytoreductive surgery, indocyanine green, fluorescence

## Abstract

**Simple Summary:**

Indocyanine green (ICG) is a dye that helps surgeons see the blood supply to tissues. In this study, (MIRRORS ICG) researchers wanted to see if this dye could help find cancer in women with advanced ovarian cancer undergoing robotic surgery after chemotherapy. After injecting ICG, a special camera was used to look at the whole abdomen and pelvic area. In this study, 102 tissue samples were taken to see if ICG helped identify cancerous tissue. The results showed that ICG correctly identified cancer in 91.1% of cases but had a low specificity of 13.0%, meaning it often falsely indicated cancer. This technique did not significantly help in detecting cancer in patients with widespread disease, but it showed some potential in those who had responded well to chemotherapy and had few remaining cancer spots. Molecular imaging with targeted dyes could enhance precision surgery in the future.

**Abstract:**

Indocyanine green (ICG) is a fluorescent dye used for sentinel lymph node assessment and the assessment of perfusion in skin flaps and bowel anastomoses. ICG binds serum proteins and behaves as a macromolecule in the circulation. Tumour tissue has increased vascular permeability and reduced drainage, causing macromolecules to accumulate within it. MIRRORS ICG is designed to determine whether indocyanine green (ICG) helped identify metastatic deposits in women undergoing robotic interval cytoreductive surgery for advanced-stage (3c+) ovarian cancer. Peritoneal surfaces of the abdominal and pelvic cavity were inspected under white light and near-infrared light (da Vinci Si and Xi Firefly Fluorescence imaging, Intuitive Surgical Inc.) following intravenous injection of 20 mg ICG in sterile water. Visibly abnormal areas were excised and sent to histopathology, noting IGC positivity. In total, 102 biopsies were assessed using ICG. Intravenous ICG assessment following neoadjuvant chemotherapy had a sensitivity of 91.1% (95% CI [82.6–96.4%]), a specificity of 13.0% (95% CI [2.8–33.6%]), a positive predictive value of 78.3% (95% CI [68.4–86.2%]), and a negative predictive value of 30.0% (95% CI [6.7–65.2%]) False-positive samples were seen in 9/20 patients. Psammoma bodies were noted in the histopathology reports of seven of nine of these patients with false-positive results, indicating that a tumour had been present (chemotherapy-treated disease). This study demonstrates the appearance of metastatic peritoneal deposits during robotic cytoreductive surgery following the intravenous administration of ICG in women who have undergone neoadjuvant chemotherapy for stage 3c+ advanced ovarian cancer. A perfusion assessment using indocyanine green (ICG) peritoneal angiography during robotic interval cytoreductive surgery for advanced ovarian cancer did not clinically improve metastatic disease identification in patients with high-volume disease. The use of ICG in patients with excellent response to chemotherapy where few tumour deposits remained shows some promise. The potential of molecular imaging to enhance precision surgery and improve disease identification using the robotic platform is a novel avenue for future research.

## 1. Introduction

Indocyanine green (ICG) is a widely available intravenous fluorescent dye used for cardiac, microcirculatory, and tissue perfusion diagnostics. ICG is used for sentinel lymph node assessment in endometrial, vulval, and cervical cancers, and other therapeutic indications include an assessment of perfusion in skin flaps and bowel anastomoses. The incidence of adverse events is low (1 in 10,000–42,000) in patients [1]. ICG is not metabolised and stays within blood vessels, binding to serum proteins and behaving like a macromolecule in circulation [2]. Macromolecules accumulate in tumour tissue due to increased vascular permeability and reduced drainage. This phenomenon is called the “enhanced permeability and retention” (EPR) effect and has been observed in most solid tumours [1].

Within ophthalmology, ICG dye given intravenously is used to detect choroidal neovascularisation (the formation of new blood vessels), which is seen in age-related macular degeneration. The anatomy of these new vessels is abnormal and characterised by their large size with varying diameters and convolutedness [3]. Peritoneal inflammation and cancer invasion cause neovascularisation, with abnormal blood vessel patterns seen over the surface of peritoneal metastatic deposits [3]. ICG dye is generally very well tolerated but it does have a very small risk of severe allergic reaction. This affects fewer than 1 in 10,000 patients [2].

The MIRRORS study was recently completed, and the results published by our team at Royal Surrey NHS Foundation Trust in Guildford, UK [4]. MIRRORS (***M****inimally **I**nvasive **R**obotic surgery*, ***R****ole in optimal debulking **O**varian cancer*, ***R****ecovery & **S**urvival*) was a prospective cohort feasibility study of robotic interval debulking surgery (IDS) in women with advanced-stage epithelial ovarian cancer. The objective of MIRRORS was to establish the feasibility and safety of a randomised controlled trial (RCT) of robotic interval debulking surgery (IDS) for epithelial ovarian cancer using the MIRRORS protocol [5]. MIRRORS recruited women with stage 3c+ epithelial ovarian cancer undergoing neoadjuvant chemotherapy who had a pelvic mass ≤ 8 cm [4,5,6]. Women who accepted recruitment underwent an initial laparoscopic assessment at the time of their surgery, followed by the decision to proceed with robotic cytoreductive surgery or open surgery (“MIRRORS Protocol”). Both approaches aimed to remove all visible disease, converting to open surgery if necessary to achieve this. MIRRORS found that robotic interval debulking surgery appears to be safe and feasible in women with a pelvic mass of ≤8 cm. Full details, including the surgical protocol and feasibility study findings, is included in our recent publication [4]. MIRRORS ICG was an ancillary study within the MIRRORS study. Only women who proceeded to robotic interval cytoreductive surgery took part in MIRRORS ICG.

Standard ovarian cancer surgery involves the careful assessment of the abdominal and pelvic cavities, identifying and removing all tumour deposits and taking biopsies. This is to remove disease burden (cytoreduction) and to stage the disease. Survival in ovarian cancer is strongly associated with the ability to remove all visible disease and tumour sensitivity to platinum-based chemotherapy [7].

The aim of MIRRORS ICG was to determine whether intravenous Indocyanine green (ICG) assisted in the identification of metastatic deposits in women undergoing robotic interval cytoreductive surgery for advanced-stage ovarian cancer.

## 2. Methods

### 2.1. Objective

The objective of this study was to determine whether intravenous indocyanine green (ICG) assisted in the identification of metastatic deposits in women undergoing robotic interval cytoreductive surgery for advanced-stage ovarian cancer.

### 2.2. Study Endpoints

The endpoint was considered the detection of abnormal blood vessel patterns (known as neovascularisation) within peritoneum by looking for the pooling of ICG and correlation with histological diagnosis of deposits showing ICG pooling (ICG was not used to guide where biopsies were taken or whether tissue was removed).

### 2.3. Participants

#### 2.3.1. Inclusion Criteria

The criteria included women ≥18 years with stage IIIc-IVb epithelial ovarian cancer suitable for IDS with a ≤8 cm pelvic mass proceeding with robotic IDS and enrolled in the MIRRORS study.

#### 2.3.2. Exclusion Criteria

The criteria excluded women who lacked the capacity to complete trial documentation, were not medically fit for laparoscopy (e.g., severe aortic stenosis) or where required specialist surgical support recommended open surgery. Additionally, women with severe renal insufficiency GFR < 55 mL/min, known allergy to iodine or ICG, and hyperthyroidism were excluded.

### 2.4. Setting

MIRRORS ICG was based at Royal Surrey NHS Foundation Trust, UK, a gynaecological cancer centre and Intuitive Robotic Epicentre. Recruitment occurred between 26 June 2020 and 21 May 2021.

### 2.5. Statistical Analysis

As a feasibility study, emphasis was placed on descriptive statistics for patient demographic data. Study data were analysed in Microsoft Excel 2016. Sensitivity, specificity, and positive and negative predictive values were calculated as described by Monaghan et al. [8].

### 2.6. Trial Design: Interventional Observational Diagnostic Study

The Alexis laparoscopic system ring retractor (Applied Medical, Rancho Santa Margarita, CA, USA) was used to perform a 3–4 cm open entry. An initial assessment was completed using the Da Vinci robot laparoscopic camera (Intuitive Surgical, Sunnyvale, CA, USA); following this, surgery proceeded immediately to robotic or open IDS (MIRRORS-protocol). Robotic surgery was performed using the Da Vinci Si and Xi system using the Intuitive Firefly fluorescence imaging system (Intuitive Surgical, Sunnyvale, CA, USA). Patients identified as suitable for robotic IDS as part of the MIRRORS study were included in MIRRORS ICG.

Peritoneal surfaces of the abdominal and pelvic cavity were examined for tumour deposits under white and near-infrared light, using the da Vinci Si and Xi Firefly Fluorescence imaging system (Intuitive Surgical Inc.) following intravenous injection of 20 mg of ICG (5 mg/mL sterile water) [9]. Only visibly abnormal areas suspicious of disease were removed and sent to histopathology as per the study protocol and ethics agreement. Wherever possible, and without disrupting the ongoing surgical procedure, specimens were excised, identified, and labelled as being ICG+ve or negative. Histology reports were cross-referenced with the surgical findings of biopsies being ICG-positive or negative, and sensitivity, specificity, and positive and negative predictive values were calculated. Figure 1 presents the chemical structure of ICG and trial methodology [2].

### 2.7. Trial Registration

The study was registered prior to the first patient recruitment at ClinicalTrials.gov: NCT04402333 (https://clinicaltrials.gov/ct2/show/study/NCT04402333 (accessed 17 July 2024). Concerning ethics approval and consent to participate, MIRRORS/MIRRORS ICG were approved by the London Riverside Research Ethics Committee (Ref: 20/LO/0262 IRAS project ID: 261933). MIRRORS was performed in accordance with the Declaration of Helsinki. Written informed consent was obtained from all participants prior to them taking part.

## 3. Results

In total, 20/23 patients proceeded to robotic interval cytoreductive surgery in the MIRRORS study. These patients were included in the MIRRORS ICG study. Table 1 and Table 2 provide a breakdown of participant demographics, both patient- and disease-related. All women had high-grade serous tumours and were BRCA-negative and none had had a complete response to chemotherapy according to Response Evaluation Criteria in Solid Tumors version 1.1 (RECIST 1.1). Median peritoneal cancer index score was 24 (6–31) and 55% had FIGO stage 4 disease.

In the 20 women undergoing robotic interval debulking surgery as part of the MIRRORS study, an assessment using ICG was performed and 102 biopsies of excised abnormal looking tissue were obtained. In total, 92 of these biopsies were ICG-positive, and 10 were negative. On histology, 72 of the ICG-positive samples were found to contain viable tumour, and 20 contained none (false positive). Of the ICG-negative samples, three contained no evidence of cancer but seven were found to have viable tumour deposits (false negative). Overall, intravenous ICG assessment following neoadjuvant chemotherapy had a sensitivity of 91.1% (95% CI [82.6–96.4%]), a specificity of 13.0% (95% CI [2.8–33.6%]), a positive predictive value of 78.3% (95% CI [68.4–86.2%]), and a negative predictive value of 30.0% (95% CI [6.7–65.2%]). The findings are summarised in Table 3.

Within two minutes following intravenous injection of ICG, fluorescence was diffusely visible throughout the abdomen, particularly concentrated within the small bowel serosa and the liver; at this point, it was difficult to identify discrete lesions due to the generalised brightness of the ICG. Concentrated pooling around tumour deposits, as illustrated in Figure 2 and Figure 3, occurred approximately 45 min later, making identification easier. In women with extensive peritoneal disease, the use of intravenous ICG did not improve the identification of tumour deposits, as almost the entire surface of the peritoneum was positive for ICG and appeared green. Disease over the right hemi-diaphragm remained poorly visualised due to the brightness of the liver under near-infrared light.

The pooling of ICG around tumour on the fallopian tube can be seen in Figure 2B. In Figure 2D, an ICG-negative small bowel mesenteric nodule is associated with surrounding neovascularisation and ICG pooling. Similarly, increased ICG pooling is seen within the adhesions surrounding the appendix in Figure 2E. These lesions were all confirmed to have deposits of high-grade serous cancer on final histology.

Psammoma bodies were noted on histology in seven of nine women with false-positive samples. False negative results occurred in four women, all with a high burden of disease (median PCI: 26; range 23–31) Of these four women, only one achieved R = 0 (no visible residual disease). This was a case of delayed interval debulking surgery after six cycles.

## 4. Discussion

This study demonstrates the appearance of metastatic peritoneal deposits during robotic cytoreductive surgery following intravenous administration of ICG in women who have undergone neoadjuvant chemotherapy for stage 3c+ advanced ovarian cancer. Our findings indicate that ICG is not particularly helpful in patients with high-volume disease due to its high sensitivity (91.1%) and low specificity (13.0%). These same qualities may be useful in identifying tumour deposits in patients with low-volume disease. This paper provides a detailed illustration of these findings and highlights potential avenues for future research, such as the development of more targeted dyes. 

ICG is widely available and used for different applications but is not a targeted fluorescent dye. ICG is known and was noted to pool around areas of trauma, inflammation, or fibrosis, without any tumour found on histology. Areas of fibrosis may be the result of chemotherapy-treated disease, particularly in those who were found to have Psammoma bodies noted on histology (seven of nine patients with false-positive samples). Overall, ICG had poor specificity (13.0%) in identifying truly negative samples and did not appear to improve tumour deposit identification in women with a high burden of disease undergoing robotic-assisted interval debulking surgery. These results may be skewed by the higher prevalence of ICG-positive samples examined. As an observational study, the study protocol dictated that only tissue that was visibly abnormal was excised. The outcome of this was that the majority of samples taken were ICG-positive.

### 4.1. Results in the Context of the Published Literature

Publications on the use of ICG to detect peritoneal metastasis are sparse, but initial studies have been reported in hepatocellular carcinoma, colorectal cancer (both in vivo and ex vivo), and gastric cancer in a mouse model using an ICG-labelled antibody [10,11,12,13,14].

Tummers et al. (2015) observed the effects of intravenous ICG in 10 patients undergoing primary laparotomy surgery for diagnosis and staging for suspected ovarian cancer [15]. Of these ten patients, six had malignant disease of the ovary/fallopian tube, with only two having disease outside of the ovary. Tummers et al. (2015) found that in these two women, all the metastatic deposits fluoresced under near-infrared light, but in the patients with benign disease, non-malignant lesions were also noted to fluoresce. In a population of patients suspicious but not known to have cancer, ICG did not distinguish between malignant, reactive, or benign disease.

Veys et al. (2017) evaluated the use of intravenous ICG at laparotomy for both primary and interval debulking surgery [16]. Intraoperative assessments of scar tissue and ex vivo assessments of peritoneal lesions were performed. In 15 patients who had had neoadjuvant chemotherapy, 8 were noted to have scarring tissue. Furthermore, 25 incidences of peritoneal residual scar tissue were detected and imaged intraoperatively in these eight patients. 18 of these scars were ICG-positive and 7 were ICG-negative, with 14 found to be benign and 11 malignant at histopathology. Veys et al. found the sensitivity to be 72.7%, specificity to be 57.1%, and the positive predictive value to be 57.1% for detecting tumour cells in scars intraoperatively.

The ex vivo assessment of 108 peritoneal lesions by Veys et al. (peritoneal nodules and residual scar tissue, 73 malignant and 35 benign) was found to have a sensitivity of 72.6%, a specificity of 54.2%, and a positive predictive value of 76.8% when using IV ICG. The majority of patients in this study had moderate peritoneal disease, with a median PCI score of 11.8 and four with a PCI score > 17 (range: 3–34). Veys et al. concluded that the use of intraoperative ICG was not able to detect additional peritoneal metastases largely due to the extent of peritoneal disease. Neither was it able to discriminate between benign and malignant “scars” following neoadjuvant chemotherapy [16].

In neurosurgery, a randomised controlled trial by Stummer et al. (2006) found that the use of fluorescence-guided surgery with 5-aminolevulinic acid led to almost a doubling of complete resections (65% vs. 36%) associated with longer progression-free survival (41% vs. 21.1%) than conventional microsurgery with white light with comparable surgical morbidity [17].

### 4.2. Strengths and Weaknesses

MIRRORS ICG formed a small part of the main MIRRORS study, with women undergoing often long and complex robotic interval debulking procedures. It was not practical to assess every single histology sample for ICG positivity. This limitation has also been noted in other studies [11,16]. Using the da Vinci Firefly Fluorescence imaging system, white light and fluorescence imaging cannot be visualised simultaneously, in contrast to other endoscopic systems [18]. The need to switch between views interrupted the surgical flow. Whilst the Firefly system is excellent for clearly defining positive lymph nodes or lymph channels from surrounding tissue, operating and resecting disease under the predominantly greyscale view is suboptimal. The development of ICG visualisation technology and its utilisation within surgical robotic systems is an exciting area for future research.

As an observational study, MIRRORS ICG was subject to potential observer bias. Masking was not possible as both surgeon and assistant were aware of the nature of the surgery and research aims. Each sample was assessed as either positive or negative by the agreement of at least two observers (the surgeon, the assistant, and a scrub nurse). Standardised procedures were in place, as per the protocol and data collection sheet.

### 4.3. Future Directions

Targeted dyes are currently under development to solve issues relating to the identification of a target that is both sensitive and specific to live peritoneal tumour deposits [19]. The ability to provide a numerical reading of the strength of the positivity of each sample intraoperatively would be of use. Current non-robotic systems do provide a heat map assessment of the uptake of ICG [18]. Further development of this in the future is of interest.

The high sensitivity of ICG showed promise. The poor specificity limits its application, particularly in patients with extensive disease and where extensive areas of ICG positivity are evident. ICG may prove more useful in patients with excellent response to chemotherapy where few tumour deposits remain. Delaying the assessment of peritoneal deposits (by proceeding with pelvic clearance/omentectomy first prior to widespread peritoneal stripping) for 45 min may enable deposits that are initially not evident to become more noticeable as the ICG pools and concentrates around lesions over time.

Minimally invasive surgery aims to provide the same surgery with improved recovery and reduced morbidity. The ability to accurately assess disease before and following cytoreductive surgery is an important field of study. Targeted fluorescent dyes could not only improve the detection of disease, and so potentially aid in the completeness of cytoreductive surgery, but also help to provide an objective record of the completeness of cytoreduction in the future.

## 5. Conclusions

This study demonstrates the appearance of metastatic peritoneal deposits during robotic interval cytoreductive surgery following the intravenous administration of ICG in women who have undergone neoadjuvant chemotherapy for stage 3c+ advanced ovarian cancer. Perfusion assessment using indocyanine green (ICG) peritoneal angiography during robotic interval cytoreductive surgery for advanced ovarian cancer did not clinically improve metastatic disease identification in patients with high-volume disease. The use of ICG in patients with excellent response to chemotherapy where few tumour deposits remained shows some promise. These cases may provide the opportunity to visualise small discrete lesions that may otherwise have been missed. ICG-positive but histology-negative biopsies may indicate chemotherapy-treated disease, given the histopathological findings of psammoma bodies indicating that a tumour had been present. This paper provides a detailed illustration of these findings and highlights potential avenues for future research. The potential of molecular imaging to enhance precision surgery and improve disease identification using a robotic platform is an exciting area for future research.

## Figures and Tables

**Figure 1 cancers-16-02689-f001:**
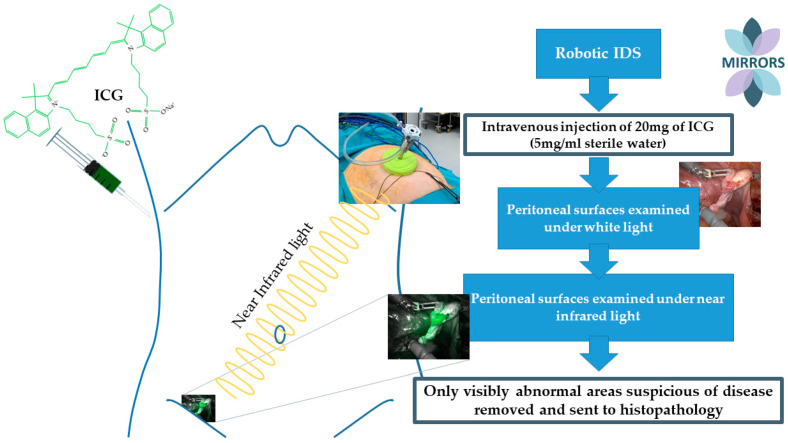
Chemical structure of indocyanine green (ICG) and trial methodology. The formation of new blood vessels, known as “neovascularisation” is seen around metastatic peritoneal deposits. The anatomy of these new blood vessels is abnormal, demonstrating large size and varying diameters and convolutedness. Intravenous ICG binds to plasma proteins in the bloodstream and behaves as a macromolecule. ICG pools in tumour tissue due to increased vascular permeability and reduced drainage in surrounding abnormal vessels. Here, tumour deposits are highlighted by ICG, on the right fallopian tube.

**Figure 2 cancers-16-02689-f002:**
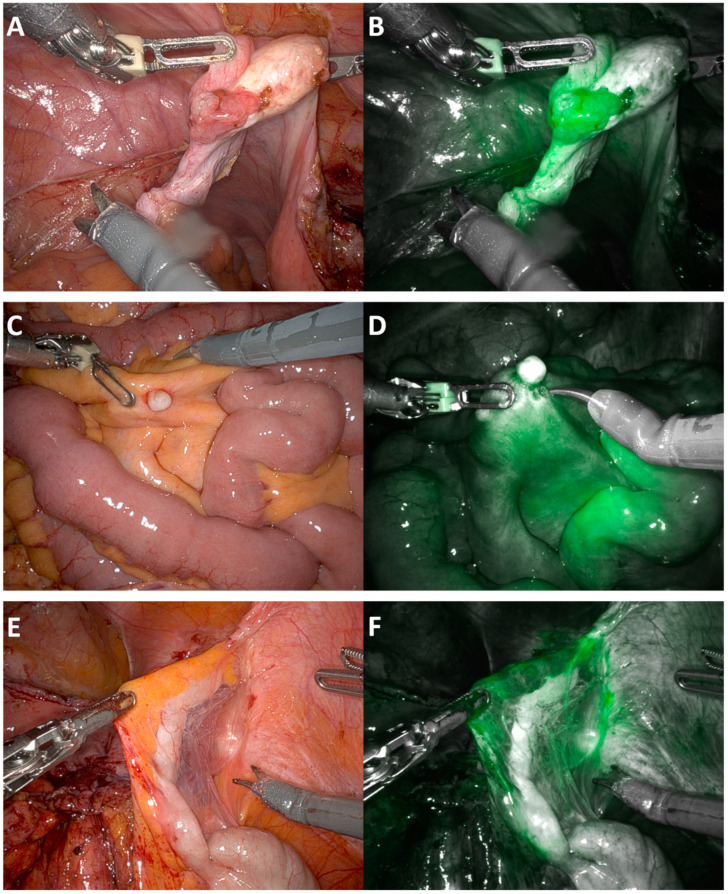
Peritoneal angiography appearance of peritoneal disease under white and near-infrared light. (**A**) Fallopian tube tumour. (**B**) Fallopian tube tumour under near-infrared light. (**C**) Calcified nodule on small bowel mesentery. (**D**) Calcified nodule on small bowel mesentery under near-infrared light. (**E**) Appendix attached to pelvic side wall. (**F**) Appendix under near-infrared light. All lesions shown above confirmed to contain viable tumour on histology.

**Figure 3 cancers-16-02689-f003:**
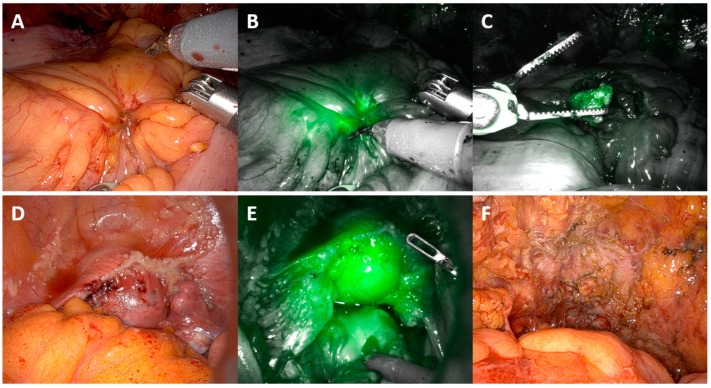
Peritoneal angiography—appearance of peritoneal disease under white and near-infrared light. (**A**) Sigmoid mesentery tumour deposit. (**B**) Sigmoid mesentery tumour deposit under near-infrared light. (**C**) Tumour deposit excised from (**B**). (**D**) Pelvic peritoneal disease overlying uterus and bladder (**E**) Pelvic peritoneal disease under near-infrared light (**F**) Appearance of pelvis following removal of uterus both tubes and ovaries and full pelvic peritoneal stripping (BMI 40). All ICG+ve lesions shown confirmed to have viable tumour on review of histology.

**Table 1 cancers-16-02689-t001:** Demographics—patient factors.

		MIRRORS ICG—Route of Surgery Robotic (n = 20)					
		Mean	Median (IQR)	Minimum	Maximum	n	n%
Age (years)		67.2	68.5 (11.5)	53.0	83.0	20	100.0%
Body Mass Index		24.7	24.0 (5.0)	15.2	38.9	20	100.0%
ECOG	0					7	35.0%
	1					12	60.0%
	2					1	5.0%
ASA	1					0	0.0%
	2					6	30.0%
	3					14	70.0%
Ethnicity	White British					18	90.0%
	Any other White background					1	5.0%
	Black Caribbean					1	5.0%
Parity (N)	0					5	25.0%
	1					2	10.0%
	2					9	45.0%
	3					2	10.0%
	4					2	10.0%
Smoking history	Ex-smoker					6	30.0%
	Never smoked					12	60.0%
	Smoker					2	10.0%
Number of previous abdominal surgeries	0					4	20.0%
	1					8	40.0%
	2					4	20.0%
	3					1	5.0%
	4					2	10.0%
	5					1	5.0%
Comorbidity							
Cardiac condition						3	15.0%
Previous VTE						3	15.0%
Anaemia						1	5.0%
Diabetes						2	10.0%
Vascular						1	5.0%
Hypertension						4	20.0%
Respiratory disease						5	25.0%
Dermatology condition						2	10.0%
Previous cancer						4	20.0%
Musculoskeletal/rheumatology						5	25.0%
Mental health						1	5.0%
Endocrine/autoimmune						6	30.0%

**Table 2 cancers-16-02689-t002:** Demographics—disease related.

		Route of Surgery Robotic					
		Mean	Median	Minimum	Maximum	n	n%
Number of cycles of chemotherapy prior to surgery		4	4	3	6	20	100.0%
Neoadjuvant chemotherapy	Carboplatin only					1	5.0%
	Combined Carboplatin and Paclitaxel					18	90.0%
	Other regimen					1	5.0%
BRCA	Negative					19	95.0%
	BRCA1					0	0.0%
	BRCA2					0	0.0%
	na					1	5%
Tumour type	Adenocarcinoma					0	0.0%
	Clear cell adenocarcinoma					0	0.0%
	MMMT					0	0.0%
	Neuroendocrine					0	0.0%
	Serous					20	100.0%
Tumour site	Endometrial					1	5%
	Ovary					9	45.0%
	Peritoneum					0	0.0%
	Tube					10	50.0%
Grade 3						20	100.0%
Stage	3c					9	45.0%
	4a					2	10.0%
	4b					9	45.0%
RECIST 1.1							
Complete response						0	0.0%
Partial response						18	90.0%
Stable disease						2	10.0%
Chemotherapy Response Score							
	0					1	5.0%
	1					10	50.0%
	2					5	25.0%
	3					4	20.0%

**Table 3 cancers-16-02689-t003:** Results. (Green colour denotes true positive or true negative and red colour false positive or false negative.)

		Actual (Based on Histology)		
	Total: 102 Biopsies	Positive (79)	Negative (23)	
Observed	ICG Positive (92)	72	20	PPV 78.3%
	ICG Negative (10)	7	3	NPV 30.0%
		Sensitivity 91.1%	Specificity 13.0%	

## Data Availability

The datasets generated from this current study will be available upon request from Miss Christina Uwins (christina.uwins@nhs.net) as raw anonymised data for up to 5 years following completion of the study. Participants gave their consent for the information collected from this study to be used to support other research in the future and to be shared anonymously with other researchers.

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
