# Peer review of "MIRRORS ICG: Perfusion Assessment Using Indocyanine Green (ICG) Peritoneal Angiography during Robotic Interval Cytoreductive Surgery for Advanced Ovarian Cancer"

_cancers, 2024, doi:10.3390/cancers16152689_

Round 1
Reviewer 1 Report
Comments and Suggestions for Authors
This work presented practical results for perfusion assessment using Indocyanine green peritoneal angiography during robotic interval cytoreductive surgery for advanced ovarian cancer. The presented images and data are valuable for publication in scientific journals. From this positive viewpoint, I may suggest publication of this work in Cancers with necessary revisions. One serious problem of this manuscript is less understandable nature. Especially, this point has to be well fixed by revisions. Please see below.
1) This manuscript presents images and data as figures and tables. Therefore, basic strategy of this work may not be understandable for non-specialist readers. At least, MIRRORS has to be explained by figures.
2) Chemical structure of Indocyanine Green and its working principles have to be also represented as a new figure.
3) In general, some paragraphs are too short and look like fragmental. Please correct this general features.
Author Response
This work presented practical results for perfusion assessment using Indocyanine green peritoneal angiography during robotic interval cytoreductive surgery for advanced ovarian cancer. The presented images and data are valuable for publication in scientific journals. From this positive viewpoint, I may suggest publication of this work in Cancers with necessary revisions. One serious problem of this manuscript is less understandable nature. Especially, this point has to be well fixed by revisions. Please see below.
We are very grateful to you for taking the time to review our paper and your valuable recommendations to improve it.
1) This manuscript presents images and data as figures and tables. Therefore, basic strategy of this work may not be understandable for non-specialist readers. At least, MIRRORS has to be explained by figures.
Many thanks for this feedback. Further detail and updated reference [4] for the MIRRORS study has been added to the introduction (page 2 paragraph 3). Additionally Figure 1 has been added to help describe the study pictorially (page 4).
2) Chemical structure of Indocyanine Green and its working principles have to be also represented as a new figure.
Many thanks for this suggestion. Figure 1 has been added to address this (page 4).
3) In general, some paragraphs are too short and look like fragmental. Please correct this general features.
Many thanks for highlighting this. I have addressed this throughout.
Reviewer 2 Report
Comments and Suggestions for Authors
From a biostat and clinical epidemiology point of view, here are some comments for the Authors
- no section has been dedicated to descriptive, inferential, modeling stats
- no table1 dedicated to main pts characteristics
- continuous covariates have to be reported as median/IQR
- poor take-home message, due to the extremely reduced sample size
Author Response
Many thanks for your feedback. Your time taken to review our paper is genuinely appreciated.
From a biostat and clinical epidemiology point of view, here are some comments for the Authors
- no section has been dedicated to descriptive, inferential, modeling stats
The following section has been added to address your point:
2.5. Statistical Analysis
As a feasibility study, emphasis was on descriptive statistics for patient demographic data. Study data was analysed in Microsoft Excel. Sensitivity, specificity, positive and negative predictive values were calculated as described by Monaghan et al.[8]
- no table1 dedicated to main pts characteristics
Many thanks for pointing out this typesetting error. This has been addressed Tables 1+2 have been moved to the appropriate point in the paper and references to them updated (page 5+6).
- continuous covariates have to be reported as median/IQR
IQR has been added to table 1 as suggested
- poor take-home message, due to the extremely reduced sample size
The conclusion has been reviewed and updated to provide a more balanced outcome
6. Conclusions
Perfusion assessment using Indocyanine green (ICG) peritoneal angiography during robotic interval cytoreductive surgery for advanced ovarian cancer did not clinically improve metastatic disease identification overall. The use of ICG in patients with excellent response to chemotherapy where few tumour deposits remain shows some promise. These cases may provide the opportunity to visualize small discrete lesions that may otherwise have been missed. The appearance of metastatic deposits following intravenous ICG during robotic surgery is of interest and gives a glimpse of future applications. ICG positive but histology negative biopsies may indicate chemotherapy treated disease given histopathological findings of psammoma bodies indicating tumour had been present. The potential of molecular imaging to enhance precision surgery and improve disease identification using the robotic platform is an exciting area for future research.
Reviewer 3 Report
Comments and Suggestions for Authors
Standard ovarian cancer surgery involves careful assessment of the abdominal and pelvic cavities, identifying and removing all tumour deposits and taking biopsies. This is to remove disease burden (cytoreduction) and to stage disease. Survival in ovarian cancer is strongly associated with the ability to remove all visible disease and tumour sensitivity to platinum based chemotherapy.
The subject of the study is topical with real interest for the future trials because we need to estimate the oncological benefits at a large scale of patients.
The introduction of the article presents originality by proposing a topic with a huge academic potential.
The bibliographic data inserted along the article presents a qualitative chronology. The subject of the article represents a true scientific revolution in its field.
The material and methods section of the article presents a quantitative and qualitative exposition of the research plan, respectively a good reproducibility in order to develop other studies with this theme.
The results of the article present a logical and chronological exposition outlining qualitative aspects of the benefit. The figures and tables keep a specific chronology throughout their exposition, presenting qualitative aspects related to the subject of the article.
Minimally invasive surgery aims to provide the same surgery with improved recovery and reduced morbidity. The ability to accurately assess disease before and followingcytoreductive surgery is an important field of study. Targeted fluorescent dyes could notnonly improve the detection of disease, and so potentially aid in the completeness of cytoreductive surgery, but also help to provide an objective record of the completeness ofmcytoreduction in the future.
The topic of the article is a real interest for the future with major importance in this field. I consider it necessary to develop new studies on this subject and implement them on a population scale. The article presents an important research point with an optimal linguistic exposition, having an exponential potential for the future. This present article is written in a clear and concise manner.
Although the article presents the characteristic limitations as you mentioned from my point of view presents originality, with an optimal literary exposition, representing a topic of real interest for the future with objective results at the research level. The article represents a launching platform in its field and from the point of view of the characteristics it is included for publication.
Author Response
Many thanks for taking the time to review our paper.
Standard ovarian cancer surgery involves careful assessment of the abdominal and pelvic cavities, identifying and removing all tumour deposits and taking biopsies. This is to remove disease burden (cytoreduction) and to stage disease. Survival in ovarian cancer is strongly associated with the ability to remove all visible disease and tumour sensitivity to platinum based chemotherapy.
The subject of the study is topical with real interest for the future trials because we need to estimate the oncological benefits at a large scale of patients.
The introduction of the article presents originality by proposing a topic with a huge academic potential.
The bibliographic data inserted along the article presents a qualitative chronology. The subject of the article represents a true scientific revolution in its field.
The material and methods section of the article presents a quantitative and qualitative exposition of the research plan, respectively a good reproducibility in order to develop other studies with this theme.
The results of the article present a logical and chronological exposition outlining qualitative aspects of the benefit. The figures and tables keep a specific chronology throughout their exposition, presenting qualitative aspects related to the subject of the article.
Minimally invasive surgery aims to provide the same surgery with improved recovery and reduced morbidity. The ability to accurately assess disease before and followingcytoreductive surgery is an important field of study. Targeted fluorescent dyes could notnonly improve the detection of disease, and so potentially aid in the completeness of cytoreductive surgery, but also help to provide an objective record of the completeness ofmcytoreduction in the future.
The topic of the article is a real interest for the future with major importance in this field. I consider it necessary to develop new studies on this subject and implement them on a population scale. The article presents an important research point with an optimal linguistic exposition, having an exponential potential for the future. This present article is written in a clear and concise manner.
Although the article presents the characteristic limitations as you mentioned from my point of view presents originality, with an optimal literary exposition, representing a topic of real interest for the future with objective results at the research level. The article represents a launching platform in its field and from the point of view of the characteristics it is included for publication.
Thank you for your thorough and thoughtful review. We are grateful for your positive feedback on the originality, importance and the quality of our work. We particularly appreciate you recognising the potential it has for significant advancements in the field. We are honoured by your kind words regarding the paper's clarity and concise writing style. Thank you again for your supportive review.
Round 2
Reviewer 2 Report
Comments and Suggestions for Authors
Most of concerns remain tovbe solved, the manuscript shows a poor take-home message and a marked lackness of novelty
Author Response
Comment 1:
Most of concerns remain tovbe solved, the manuscript shows a poor take-home message and a marked lackness of novelty
Response 1:
Many thanks for your thorough review and feedback on our paper. We acknowledge your comments. Whilst our study presents largely negative results it offers important insights into the application of ICG in this patient population; illustrating the appearance of metastatic deposits of high-grade serous ovarian cancer after neoadjuvant chemotherapy during interval cytoreductive surgery. Our findings indicate that ICG is not particularly beneficial in patients with a high-volume disease due to its high sensitivity and low specificity. However, these same qualities may be useful for patients with a low-volume disease. Our paper provides a detailed illustration of these findings and highlights potential avenues for future research, such as development of more targeted dyes. We have addressed your specific comments in the revised manuscript making further updates to the abstract, discussion and conclusion.